# Efficient immortalization of human dental pulp stem cells with expression of cell cycle regulators with the intact chromosomal condition

**Ai Orimoto**[1]*, **Seiko Kyakumoto**[2], **Takahiro Eitsuka**[3], **Kiyotaka Nakagawa**[3], **Tohru Kiyono**[4]*, **Tomokazu Fukuda**[1,5]*

1 Graduate School of Science and Engineering, Iwate University, Morioka, Iwate, Japan, 2 Division of Cellular Biosignal Sciences, Department of Biochemistry, Iwate Medical University, Shiwa-gun, Iwate, Japan, 3 Graduate School of Agricultural Science, Tohoku University, Sendai, Miyagi, Japan, 4 Division of Carcinogenesis and Prevention, National Cancer Center Research Institute, Tokyo, Japan, 5 Soft-Path Engineering Research Center (SPERC), Iwate University, Morioka, Iwate, Japan

* tkiyono@ncc.go.jp (TK); tomofukuda009@gmail.com (TF); aiorimoto@gmail.com (AO)

**Data Availability Statement:** All relevant data are within the paper and its Supporting Information files.

## Abstract

Clinical studies have recently demonstrated that autologous transplantation of mobilized dental pulp stem cells is a safe and efficacious potential therapy for pulp regeneration. However, some limitations need to be addressed, such as the high cost of the safety and quality control tests for isolated individual dental pulp cell products before transplantation. Therefore, more efficient *in vitro* culturing of human dental pulp stem cells might be useful for providing low cost and high reliability testing for pulp regeneration therapy. In this study, we established a novel immortalized dental pulp stem cell line by co-expressing a mutant cyclin-dependent kinase 4 (CDK4$^{R24C}$), Cyclin D1, and telomerase reverse transcriptase (TERT). The established cell line maintained its original diploid chromosomes and stemness characteristics and exhibited an enhanced proliferation rate. In addition, we showed the immortalized human dental pulp stem cells still keeps their osteogenic and adipogenic differentiation abilities under appropriate culture conditions even though the cell proliferation was accelerated. Taken together, our established cell lines could serve as a useful *in vitro* tool for pulp regeneration therapy, and can contribute to reproducibility and ease of cell handling, thereby saving time and costs associated with safety and quality control tests.

## Introduction

Human dental pulp stem cells exhibit high proliferation, greater tissue regeneration capabilities, lower immunogenicity, and greater plasticity than those of other mesoderm-derived mesenchymal stem cells [1]. Furthermore, unlike other mesoderm-derived mesenchymal stem cells, human dental pulp stem cells are isolated easily from extracted teeth without causing secondary damage or ethical controversy. Paino et al. have reported that human dental pulp stem

**Funding:** The authors received no specific funding for this work.

**Competing interests:** The authors have declared that no competing interests exist.

cells are a good option for applications in human bone tissue engineering without the use of scaffolds in vitro and in vivo [2]. Therefore, human dental pulp stem cells have attracted attention as candidate cells for stem cell therapy for various disorders, including the regeneration of lost pulp and dentin in the root canal space [3,4]. Recently, a pilot clinical study and a phase I clinical trial in humans have been reported that demonstrated that autologous transplantation of mobilized dental pulp stem cells is a safe and efficient therapeutic approach [5–7]. However, there are some limitations to this approach, such as the high cost of the safety and quality control tests for isolated individual dental pulp cell products before transplantation. Therefore, more effective tools are needed to provide low cost and high reliability for stem cell-mediated regeneration therapy of lost pulp.

Our research group has previously reported efficient immortalization in multiple species via co-expression of R24C mutant cyclin-dependent kinase 4 (CDK4$^{R24C}$), Cyclin D1, and telomere reverse transcriptase (TERT) [8–14]. This immortalization method using mutant CD<u>K4</u>, Cyclin <u>D</u>1, and <u>T</u>ERT was termed "K4DT" in reference to the introduced genes. The chromosomal pattern of cells established using the K4DT method is retained, along with the nature of primary cells, possibly due to the intact function of p53 [13,15–17]. We also recently demonstrated that our corneal epithelial cell line, established with the K4DT immortalization method, can be a useful tool to detect eye toxicity, and it can be used as a new resource for *in vitro* ocular toxicity testing [18]. These findings indicated that applying the K4DT immortalization method to human dental pulp stem cells might be useful in generating a more effective tool to evaluate the safety and quality of isolated individual dental pulp cell products before transplantation. We speculated that *in vitro* culturing of human dental pulp stem cells immortalized by the K4DT method might be useful as a biological resource to reduce the cost of pulp regeneration therapy.

With this aim in mind, we transduced CDK4$^{R24C}$, Cyclin D1, and TERT into human dental pulp stem cells via retrovirus. We successfully established immortalized human dental pulp stem cells and evaluated the characteristics of the cells.

## Materials and Methods

### Cell Culture

Human dental pulp stem cells (PT-5025) were purchased from Lonza Japan Ltd (Tokyo, Japan) and were cultured according to the manufacturer's instructions.

### Preparation and infection of recombinant retroviruses into human dental pulp stem cells

To immortalize primary human dental pulp stem cells, we prepared recombinant retroviruses expressing R24C mutant cyclin-dependent kinase 4 (CDK4$^{R24C}$), Cyclin D1, and TERT. PQCXIP-CDK4R24C (puromycin-resistant), pQCXIN-Cyclin D1 (G418-resistant), and pCLXSH-TERT (hygromycin B-resistant) retroviral plasmids as well as pQCXIN-EGFP (G418-resistant) as a control expressing EGFP to monitor the efficiency of infection were constructed as described previously [16]. These retroviral plasmids were co-transfected into 293T cells together with packaging plasmids, pCL-GagPol and pCMV-VSV-G-RSV-Rev, by using the lipofection method [19]. Viral fluids recovered from the transfected cells were filtered through 0.45 μm disks (Sartorius, Goettingen, Germany; product code, 17598 K). Primary human dental pulp stem cells were inoculated with individual or mixed recombinant viruses in the presence of 8 μg/mL of polybrene (hexadimethrine bromide, Sigma-Aldrich, #H9268). The cell culture medium was replaced with fresh medium one day post-inoculation followed

by selection with 1 mg/mL G418, 0.8 μg/mL puromycin and/or 40 μg/mL hygromycin according to drug resistance of each vector and its combination. One week post-selection, the obtained resistant cells were expanded into new plates. We named the retrovirus-infected human dental pulp cells WT (no treatment), EGFP (infected with EGFP-expressing retrovirus), K4D (infected with mutant CD<u>K4</u> and Cyclin <u>D</u>1), TERT (infected with TERT), and K4DT (infected with mutant CD<u>K4</u>, Cyclin <u>D</u>1, and <u>T</u>ERT) cells according to the introduced genes. The infected mass population of cells was subjected to further analysis, such as western blotting and extension of the life span.

## Population doublings

To measure the cell proliferation rates of WT, K4D, and K4DT cells, we sequentially passaged the cells. All of the cell lines were cultured in DMEM with 10% FBS containing penicillin/ streptomycin. Each cell line was initially seeded into 6-well plates at a density of $5.0 \times 10^4$ cells in triplicate. When the cell line reached confluency, the cells were dispersed using 0.05% trypsin-EDTA (Life Technologies). We recorded the total number of cells in each dish using an automatic cell counter (Thermo Fisher Scientific, Waltham, MA, USA). Cells ($5.0 \times 10^4$) were then seeded into a new dish to evaluate their growth rate by determining the population doubling level (PDL). The PDL was calculated using the following equation: PDL = log2 (A/B), where A is the number of cells harvested at each passage and B is the number of seeded cells.

## Western blotting

To extract proteins from WT, K4D, and K4DT cells, we lysed cells in a buffer containing 50 mM Tris–HCl (pH 7.4), 0.15 M NaCl, 1% Triton X-100, 2.5 mg/mL sodium deoxycholate (Wako, Osaka, Japan), and a protease inhibitor cocktail (1/200 dilution, Nacalai Tesque). Total cell lysates were separated by SDS-PAGE and then transferred to PVDF membranes (Merck, Darmstadt, Germany). After the membranes were blocked with 1% nonfat dry milk with 0.05% Tween 20, they were probed with mouse anti-human CDK4 (1:2500, #sc-56277; Santa Cruz Biotechnology, Dallas, TX, USA), Cyclin D1 (1:5000, #553; MBL, Nagoya, Japan), and α-tubulin (1:1000, #sc-32293; Santa Cruz Biotechnology) antibodies. HRP-conjugated goat anti-mouse IgG (1:2000, #330; MBL) or HRP-conjugated goat anti-rabbit IgG (1:2000, #458; MBL) was used as a secondary antibody. Immunoreactive signals were detected using an Image Quant LAS-4000 mini system (GE Healthcare, Little Chalfont, UK) with Pierce ECL Western Blotting Substrate (Thermo Fisher Scientific).

## Genomic polymerase chain reaction

Genomic DNA was extracted using a NucleoSpin Tissue kit (TaKaRa Bio, Shiga, Japan) according to the manufacturer's protocol. PCR amplification was performed using Go Taq Green Master Mix (Promega, Madison, USA) based on the protocol provided by the manufacturer (2 min of pre-denaturation at 95˚C, 30 cycles of 15 sec at 95˚C, 15 sec at 55˚C, and 1 min at 72˚C). For the detection of the *Cyclin D1* cassette, the R24C mutant *CDK4* cassette, and the *TERT* cassette, the PCR primer sequences used were described in our previous publication [14]. Tuberous sclerosis type II (TSC2) was used as an internal control since the TSC2 gene is a unique gene in the human genome, and furthermore there are no TSC2 pseudogenes. For internal control of genomic amplification, we designed forward and reverse primers for TSC2, (5'-ACT AGG CAG CAA CCA GCG TCAC -3') and (5'- TGG ACC CCA TCT CGG CAC CA-3'), respectively. PCR products were separated by 1% agarose gel electrophoresis and stained with ethidium bromide.

## Cell cycle assay

The cell cycle phase was analyzed using a Muse Cell Cycle Assay Kit (#MCH100106; Merck) and Muse Cell Analyzer (#0500–3115; Merck). The detailed method for the analysis was based on the protocol provided by the manufacturer. Statistical significance was evaluated using a non-parametric Steel-Dwass test with six samples for each experimental group.

## Senescence-associated β-galactosidase staining

At passage 7, WT, K4D, and K4DT cells were seeded at a density of $5.0 \times 10^4$ cells/well in 6-well plates. After 4 days of culture, they were fixed and stained using a Senescence Detection Kit (#K320-250; Biovision, Inc., Milpitas, CA, USA) according to the manufacturer's protocol.

## Karyotype analysis

Karyotype analyses were carried out in the K4DT cells. Standard G-banding chromosome analysis was performed in the Nihon Gene Research Laboratories, Inc. (Sendai, Japan). The chromosome number was determined from 50 mitotic cells, and the detailed chromosomal condition was evaluated using G-banding in 20 mitotic cells.

## Flow cytometric analysis

Cells ($1.0 \times 10^6$ cells) were suspended in PBS containing 2% FBS and 2 mM EDTA, and stained with FITC conjugated antibodies against human CD90, CD45, and CD34 (BioLegend, San Diego, CA, USA) to characterize the K4DT cells compared with the WT cells. FITC mouse IgG1, κ Isotype control antibody and FITC mouse IgG2a, κ Isotype control antibody (BioLegend, San Diego, CA, USA) were used for the negative control experiments. Data were analyzed with a CytoFlex flow cytometry system (Beckman Coulter, Brea, CA, USA).

## The inductions and detections of osteogenic or adipogenic differentiation

To induce osteogenic differentiation, confluent cells were incubated in osteogenic induction medium consisting of MEM alpha (Thermo Fisher, 32571036) supplemented with 10% FBS, 0.01 μM dexamethasone (Sigma, D4902), 6 mM β-glycerol phosphate (Sigma, G9422), and 50 μg/ml ascorbic acid (Sigma, A7506) for 2–4 weeks. The induction medium was changed every 3 days. Bone matrix mineralization was assessed by staining using a calcified nodule staining kit with Alizarin red S (Cosmo Bio, Tokyo, Japan) according to the manufacturers' protocols. For the detection of osteogenic differentiation, we performed immunofluorescence staining with an antibody against osteocalcin (Santa Cruz, sc-390877). Primary antibodies were detected by Alexa568 (ThermoFisher Scientific) conjugated secondary antibody. Nuclear staining was performed with DAPI. Fluorescence images were obtained using a fluorescence microscope, BZ-X810 (Keyence, Osaka, Japan). To induce adipogenic differentiation, cells were cultured to 80% confluence and cultured in adipogenic induction medium consisting of high glucose DMEM (Wako) supplemented with 10% FBS, 10 μg/ml insulin (Sigma, I9278), 1 μM dexamethasone (Sigma), 0.5 mM isobutyl-methylxanthine (Sigma, 410957), and 10 μg/ml indomethacin (Sigma, 17378) for 23 days. The induction medium was changed every 3 days. Adipogenesis was assessed by staining with Oil Red O (Sigma, O0625). We captured the staining images with a microscope, BZ-8000 (Keyence).

## Results

### Establishment of cell lines of human dental pulp stem cells transduced with CDK4$^{R24C}$, Cyclin D1, and TERT

We transduced EGFP or human CDK4, Cyclin D1, and TERT into primary human dental pulp stem cells using retroviruses. We monitored the gene delivery efficiency of the retroviruses in the dental pulp stem cells using the recombinant retrovirus QCXIN-EGFP as a control. We estimated that the transduction of these genes into the dental pulp stem cells by retroviruses had an efficiency of approximately 50% (Fig 1, middle panels). Due to the presence of a neomycin-resistance gene downstream of QCXIN-EGFP, we performed selection using G418 antibiotics to purify the EGFP-expressing cells. The surviving cells selected by treatment with 1 mg/mL G418 were almost all positive for EGFP expression (Fig 1, lower panels), indicating that antibiotic selection worked properly. To evaluate the potential toxicity of these genes in human dental pulp cells, we compared the cell morphologies of the recombinant cells with those of primary cells. Although the recombinant cells' sizes were relatively smaller in K4D, K4DT, and TERT cells, the recombinant cells had a similar morphology to WT cells (Fig 2), indicating that transduction of human CDK4$^{R24C}$, Cyclin D1, and TERT via retroviruses did not cause any toxicity in these cells.

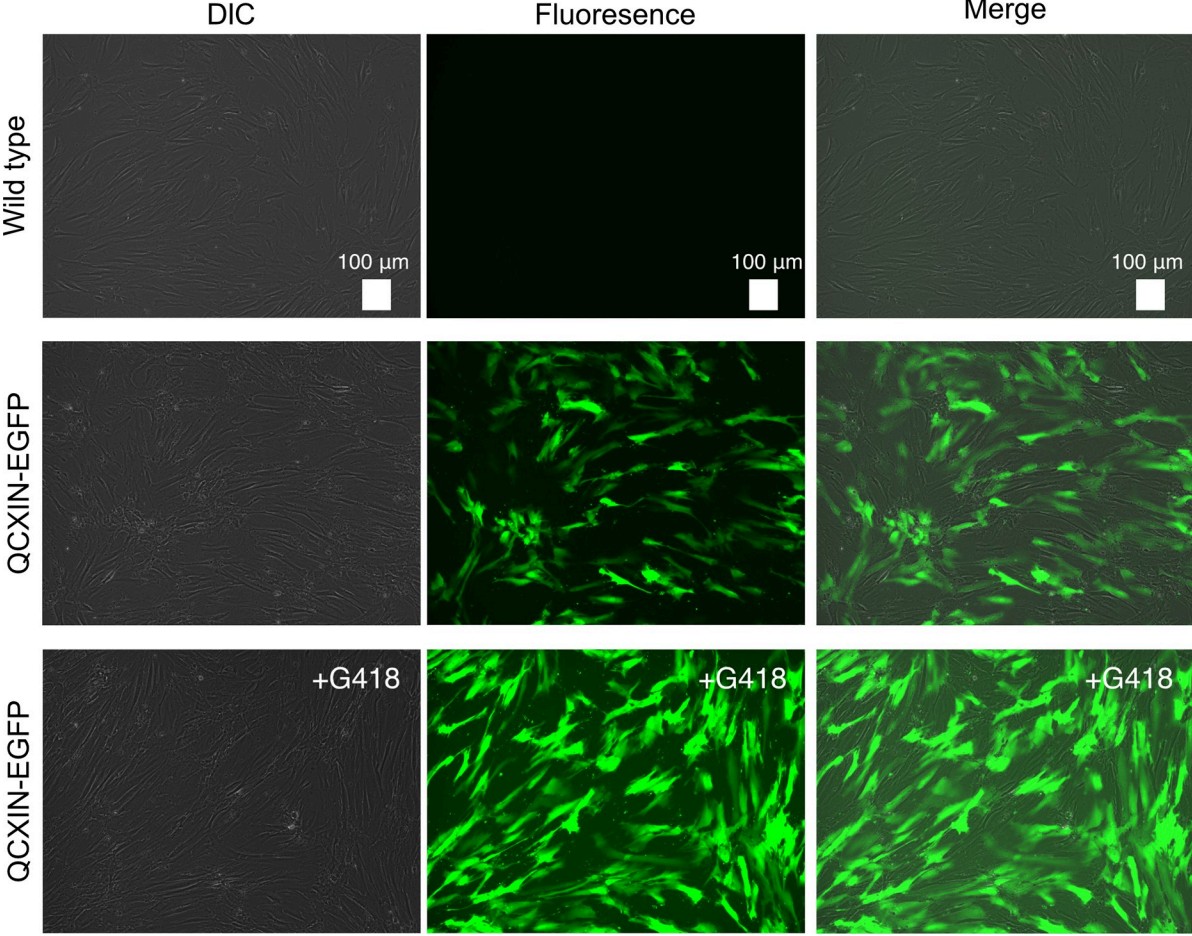

**Fig 1. Detection of fluorescence in human dental pulp stem cells expressing QCXIN-EGFP.** Differential interference contrast (DIC), fluorescence, and merged images of wild type human dental pulp cells as a control (no infection) (upper panel), QCXIN-EGFP infected human dental pulp cells (middle panels), and the surviving QCXIN-EGFP-infected human dental pulp cells selected by administration of 1000 μg/ml G418 (lower panels). Scale bar, 100 μm.

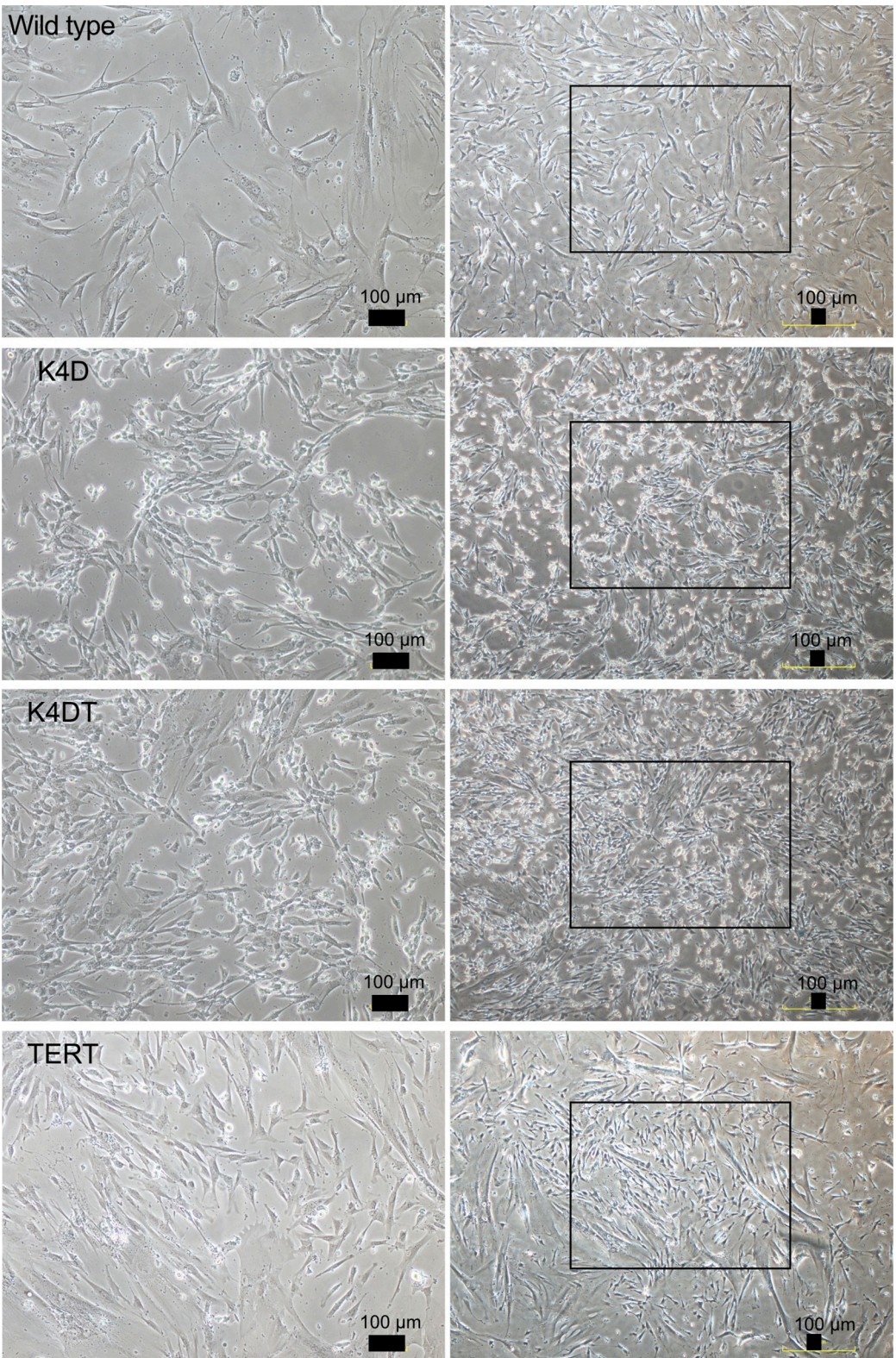

**Fig 2. The morphologies of wild type, K4D, K4DT, and TERT cells.** Left panels show high magnification of the boxed regions in the right panels. Scale bar, 100 μm.

## Detection of the genomic expression cassettes with PCR and detection of protein expression

To monitor the genomic insertion of exogenous genes, we performed PCR using genomic DNA extracted from EGFP, WT, K4D, K4DT, and TERT cells. The amplification products specific to the Cyclin D1, CDK4, and TERT expression cassettes were detected with expected ratios (Fig 3A), indicating that the expression cassettes were integrated into the cells. The results of TSC2 PCR amplification as a control showed that all genomic DNA produced sufficient amplification products, indicating that the recovery of genomic DNA and the amplification reaction worked properly. Furthermore, we carried out western blotting to detect the expression of Cyclin D1 and CDK4 proteins with specific antibodies. K4D and K4DT cells showed specific bands at the expected molecular weight, whereas WT showed a weak signal, which could be attributed to endogenous dental pulp stem cell-derived Cyclin D1 and CDK4 proteins (Fig 3B). Taken together, these data showed that the exogenous genes were inserted into the genomic DNA of human dental pulp stem cells and that functional proteins were produced by retroviral transduction.

## K4DT cells continued to proliferate without cellular senescence

We compared proliferation of the established cells with or without transgenes. At the beginning, a specific number ($5 \times 10^4$ cells/35 mm diameter dish) of WT, K4D, and K4DT cells were seeded. The K4D and K4DT cells showed an increase in cell growth speed compared to WT cells (Fig 4A). The K4DT cells did not exhibit a decrease in cell proliferation, whereas K4D cells showed slower rates of proliferation at around passage 4. Furthermore, the WT and K4D cells showed enlarged cytoplasm, and the cells stained positively for senescence-associated beta-galactosidase (SA-β-Gal) at passage 7 (Fig 4B, upper and middle panels), indicating the presence of cellular senescence in WT and K4D cells. However, the K4DT cells showed no morphological changes and did not stain positively for SA-β-Gal (Fig 4B, lower panel). These results indicate that, although the expression of CDK4$^{R24C}$ and Cyclin D1 induces dramatically enhanced cell proliferation speed until senescence at an early passage, co-expression of these genes is not enough to immortalize human dental pulp stem cells. We therefore concluded that the combined expression of CDK4$^{R24C}$, Cyclin D1, and TERT allowed us to efficiently establish immortalized cells from human dental pulp cells.

## Cell cycle assay

To compare the cell cycle distribution of cells, we performed cell cycle analysis of WT, K4D, and K4DT cells at passage 2. We compared the percentages of cells in each phase of the cell cycle. K4D and K4DT cells showed similar cell cycle profiles as WT cells, with a small fraction of S phase cells (Fig 5A). We also used six samples from each group to evaluate the potential effect of cell cycle stages. Both K4D and K4DT cells showed a significant decrease in the ratio of G0/G1 phase and an increase in the ratio of G2/M phase compared to WT cells (Fig 5B). These results indicated that cell proliferation was accelerated by the expression of mutant CDK4 and Cyclin D1.

## Karyotype analysis

We evaluated the chromosomal karyotype of K4DT cells. To obtain the chromosome number, we evaluated 50 mitotic cells. The results showed that K4DT cells displayed a chromosome number of 2n = 46, indicating that K4DT cells retained the original number of chromosomes (Fig 6A, 6B and 6C). In G-banding analysis, two samples out of 20 mitotic cells exhibited a

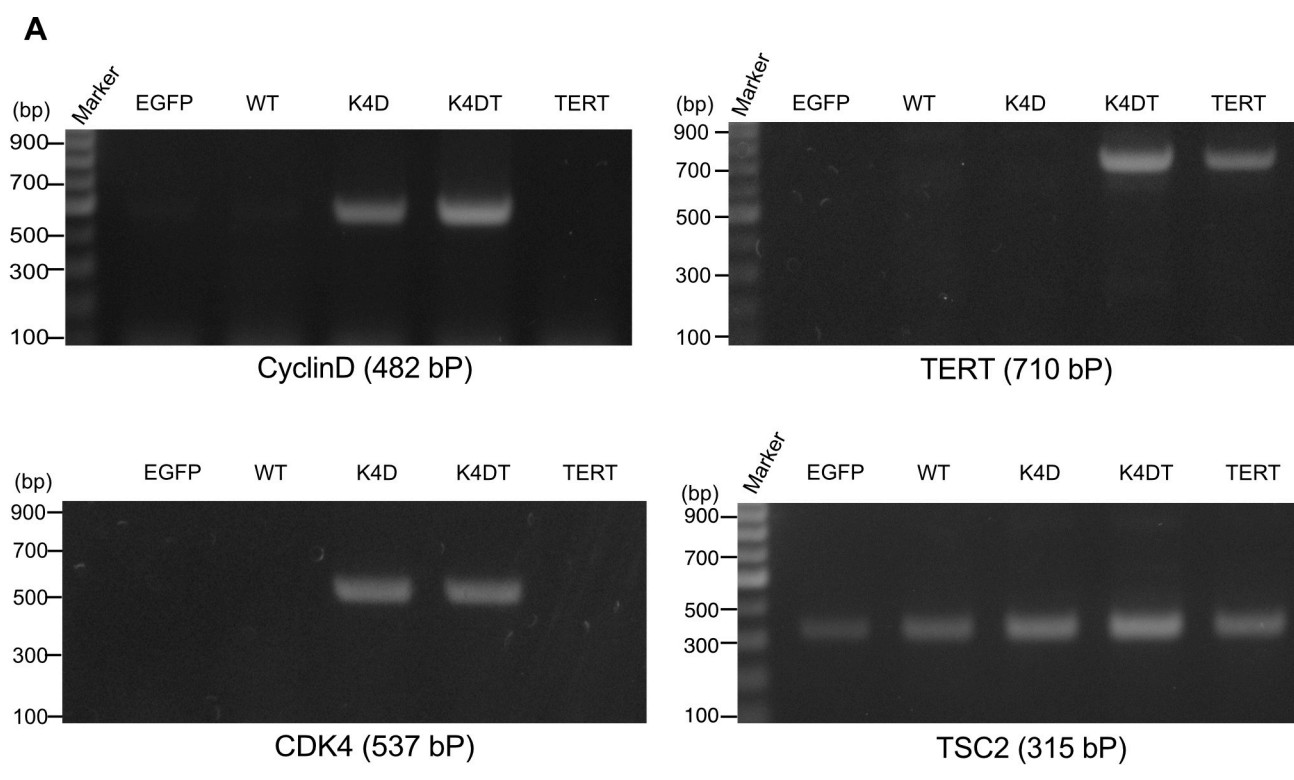

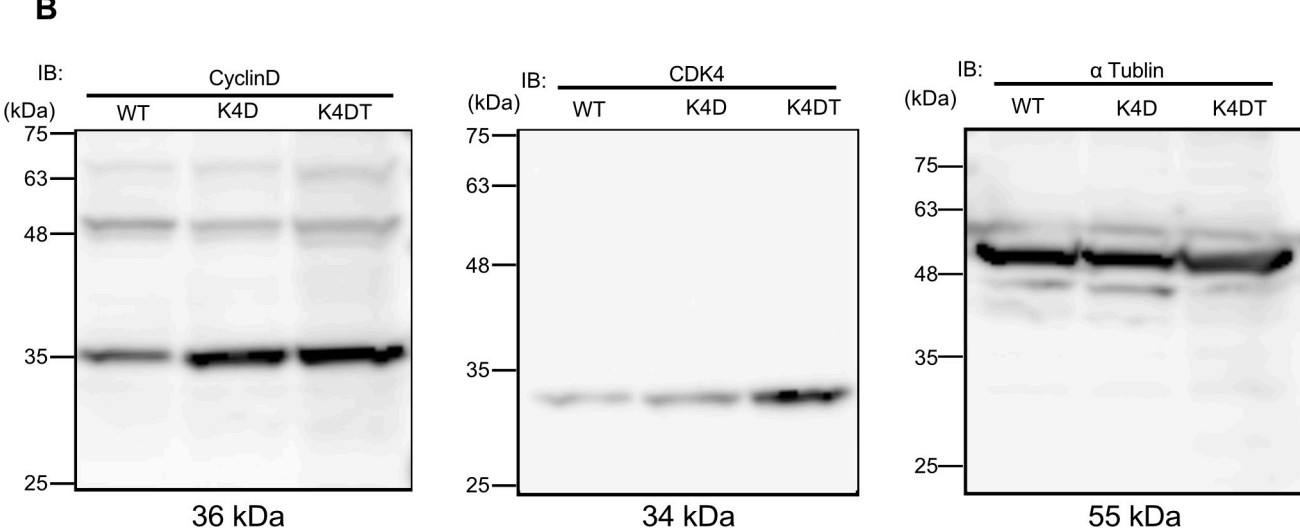

**Fig 3. Detection of genomic cassette and protein expression in the retrovirus-infected human dental pulp cells.** (A) PCR detection of CDK4, Cyclin D1, and TERT expression cassettes in the genomic DNA of EGFP, wild type (WT), K4D, and K4DT cells. PCR products from Tuberous sclerosis type 2 (TSC2) were used as an internal control. (B) Western blot analysis of wild type, K4D, and K4DT cells. The results obtained from anti-Cyclin D1, anti-CDK4, and anti-αtubulin antibody staining are shown.

chromosomal abnormality, indicating that more than 90% of K4DT cell samples had normal G banding patterns of human chromosomes (Fig 6D).

**A**

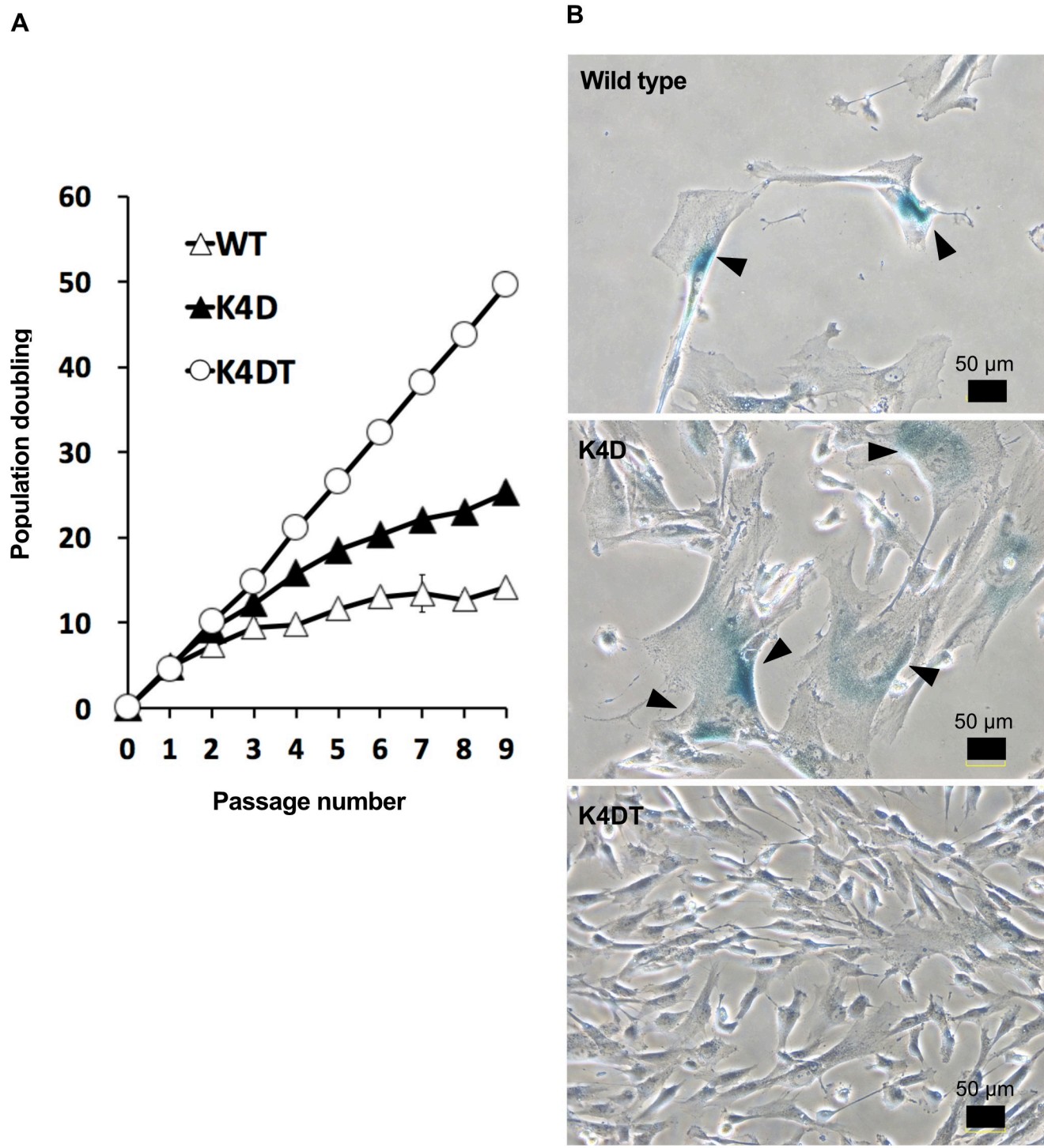

**Fig 4. Growth curve of the cell proliferation of wild type, K4D, K4DT cells.** (A) Cell growth and sequential passaging of wild type, K4D, and K4DT cells. Cell growth is represented by the cumulative population doubling value. (B) Detection of senescent cells in wild type, K4D, and K4DT cells at passage 7 by SA-beta-Gal staining. The arrows indicate positive blue staining, denoting cellular senescence. Scale bar, 50 μm.

**A**

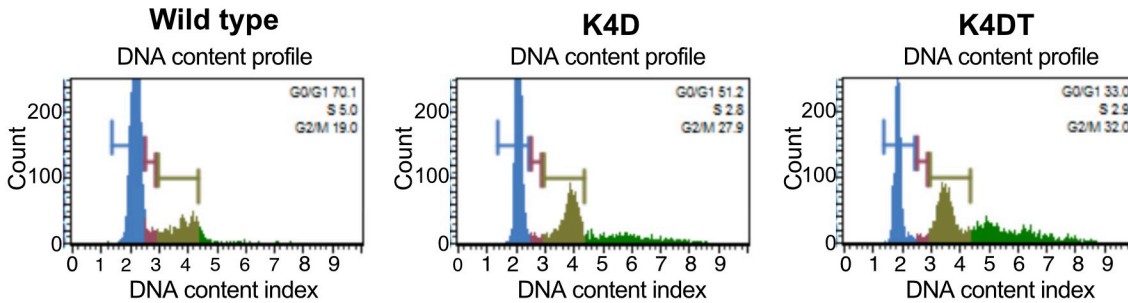

**B**

| Cell Name | Phase of cell cycle (% ± SE) | | |
|---|---|---|---|
| | G0/G1 | S | G2/M |
| Wild type | 65.4 ± 2.2 | 4.8 ± 0.2 | 21.7 ± 0.9 |
| K4D | 45.7 ± 2.7* | 3.2 ± 0.2* | 27.4 ± 0.6* |
| K4DT | 32.8 ± 0.7* | 5.2 ± 0.6 | 31.8 ± 0.3* |

**Fig 5. Cell cycle analysis of wild type, K4D, K4DT cells.** (A) Cell cycle histogram of representative results obtained from wild type, K4D, and K4DT cells using the Muse Cell Cycle kit and the Muse Cell Analyzer. (B) The representative data of each experimental group were listed. Data are presented as the mean ± standard error of the ratio of each cell cycle stage (n = 6). We used the Steel-Dwass method. $^*$p < 0.05, $^{**}$p < 0.01.

### Characterization of immortalized human dental pulp stem cells

To determine whether our immortalized cells keep the original characteristic of dental pulp stem cells, the stemness and their differentiation abilities were assessed by using the K4DT cells at passage 8, 15, or 16. Flow cytometry analysis of the K4DT cells showed more than 70% expressed CD90, which is cell surface markers in mesenchymal stem cells, whereas less than 1.0% of the K4DT expressed CD45 or CD34, which is hematopoietic cell surface markers. The WT cells showed similar results that CD90 expression was positive, whereas CD45 and CD34 expression were negative (Fig 7). These data indicated that the K4DT cells keep the original cell surface markers. In addition, K4DT cells at passages 15 showed positive staining of calcified materials by Alizarin red S after 17 days of culture in the osteoinduction medium (Fig 8A). Furthermore, we performed the immunofluorescence staining with anti-osteocalcin antibody. Osteocalcin was used for the mature osteoblast differentiation marker due to encode a bone specific protein synthesized by osteoblast [20]. The K4DT cells after 25 days of culture in the osteoinduction medium showed the osteocalcin-positive expression cells, while the cells of culture in the normal medium showed negative (Fig 8B). The data indicated that K4DT cells have the potency to differentiate into osteoblasts. We also observed that the K4DT cells at passages 16 developed numerous lipid droplets stained with Oil Red O at 23 days after the cells

A

B

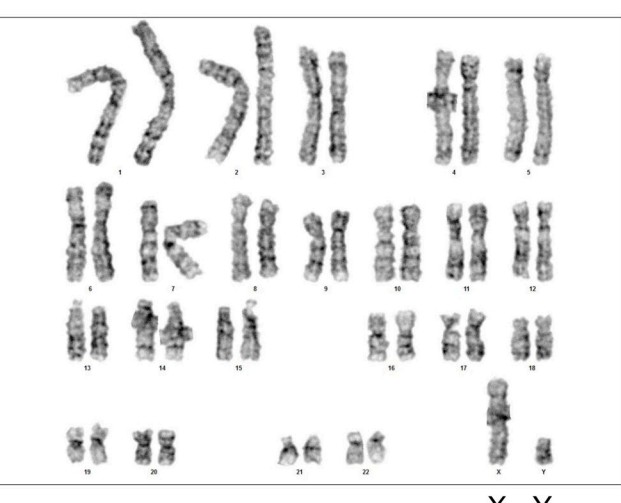

X  Y

C

| | Chromosome number | | | |
|---|---|---|---|---|
| | 45 | 46 | 47 | Diploid/total |
| K4DT | 0 | 50 | 0 | 50/50 |

D

| G banding pattern | N | % |
|---|---|---|
| 46, XY | 18 | 90 |
| 46, XY, add (2) (q21), add (7) (q22) | 1 | 5 |
| 46, XY, +5, -10, -18, +mar | 1 | 5 |

**Fig 6. Karyotype analysis of immortalized human dental pulp stem cells, termed K4DT cells.** (A) Metaphase chromosomes visualized by Giemsa staining. (B) Aligned chromosomes from K4DT cells. Sex chromosomes are indicated as X and Y. (C) Representative results of karyotype analysis of K4DT cells. (D) The representative percentage and number of metaphase chromosomes in K4DT cells by G banding analysis.

were induced to differentiate in the adipoinduction medium (Fig 9). The data indicate that the K4DT cells have the differentiation ability to adipocytes. From these results, we concluded that

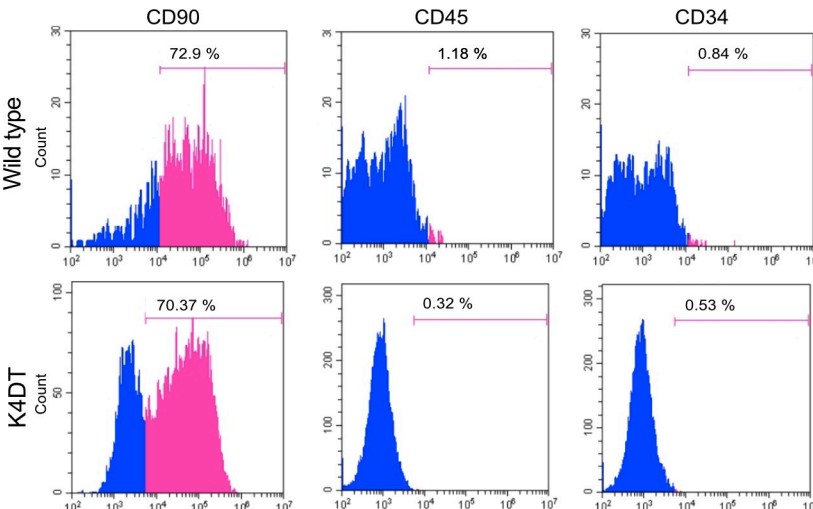

**Fig 7. Cell surface phenotype of wild type and K4DT cells by flow cytometric analysis.** The wild type and K4DT cells were positive for mesenchymal stem cell markers (CD90) and negative for hematopoietic markers (CD34 and CD45) (pink area). Isotype-identical antibodies served as the controls (blue area).

our established immortalized human dental pulp stem cells preserved the expression of stemness associated markers and their multi-differentiability even after passages.

## Discussion

In this study, by co-expressing mutated cyclin-dependent kinase 4 (CDK4$^{R24C}$), Cyclin D1, and telomerase reverse transcriptase (TERT), we established a novel immortalized human dental pulp stem cell line, termed K4DT cells, which do not have major chromosomal abnormalities or a transformed phenotype. Analysis of cell proliferation over sequential passages showed that K4DT cell lines had a more accelerated proliferation speed than K4D cell lines and primary human dental pulp stem cells. During these experiments, K4DT cells showed continuous proliferation without affecting differentiation potential and signs of senescence. This result indicated that using a combination of CDK4$^{R24C}$, Cyclin D1, and TERT efficiently immortalized human dental pulp stem cells.

Cellular stress activates the p16-Rb (retinoblastoma protein) pathway [21]. The p16 protein binds the Cyclin D1-CDK4 complex and inactivates its kinase activity, which inhibits endogenous CDK4, resulting in a slowdown of cell cycle turnover [17]. Although the overexpression of wild-type CDK4 sequesters p16, CDK4 alone does not completely suppress Rb, because its kinase activity is inhibited by p16. Overexpression of mutant CDK4 (CDK4$^{R24C}$), which is not bound by p16, in cooperation with Cyclin D1 can phosphorylate and suppress Rb in the presence of high levels of p16 [17]. However, to immortalize some cell types in culture, in addition to inactivation of the p16-Rb pathway, the expression of TERT is required to prevent progressive telomere shortening [22]. Sasaki et al. showed that neither CDK4$^{R24C}$ nor Cyclin D1 alone or in combination with TERT could efficiently immortalize primary human ovarian surface epithelium cells [14]. In fact, our present study showed that combining the expression of CDK4$^{R24C}$ and Cyclin D1 is insufficient to immortalize human dental pulp stem cells, termed K4D. We therefore concluded that the combined expression of the three genes successfully immortalized human dental pulp stem cells. We believe that our established K4DT cell line has the advantage of being easy to handle, which should increase the reproducibility and save time

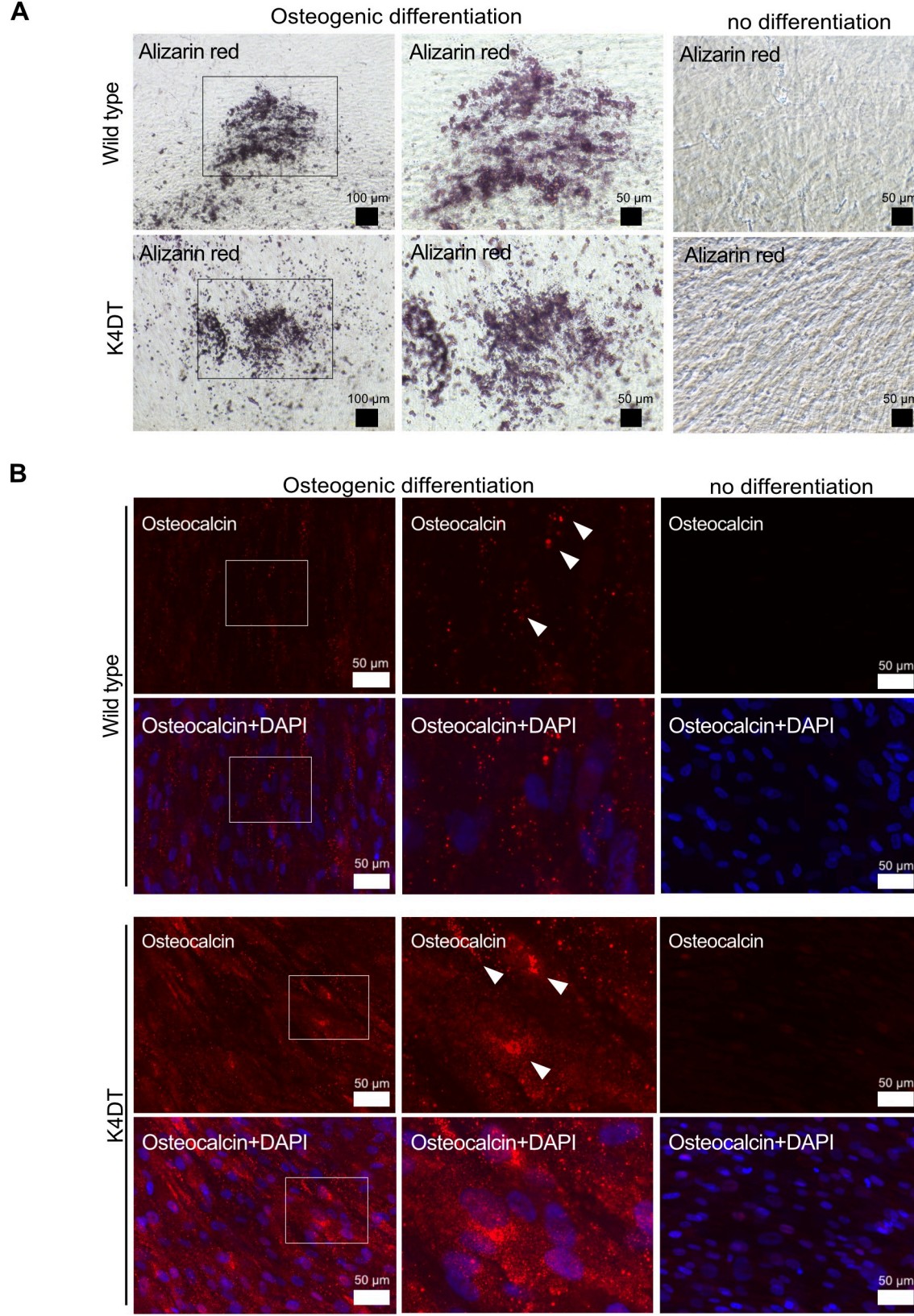

**Fig 8. Osteogenic differentiation properties of wild type and K4DT cells.** Representative Alizarin Red S staining (A) and immunostaining (B) with anti-osteocalcin antibody *(red)* and DAPI staining *(blue)* of wild type and K4DT cells. For the Alizarin Red S staining, cells were cultured in osteoinduction medium for 17 days; 25 days for the osteocalcin staining. Middle panels show high magnification of the boxed regions in the left panels. White arrows indicate the positive expression of osteocalcin.

and costs for the safety and quality control tests needed for universal dental pulp regeneration therapy.

A previous study showed that human pulp-derived cells can be immortalized with Simian Virus 40 T-antigen (SV40) [23]. Although SV40 is well known to be able to effectively immortalize most cell types by inactivating the p53 and Rb pathways and interacting with a wide range of cellular proteins [24], the expression of SV40 frequently leads to chromosomal abnormalities and polyploidy in transformed human cells [25–27]. Our established immortalized human dental pulp stem cells showed no change in chromosome number, 46+XY. These results indicated that the K4DT immortalization method, which uses co-expression of CDK4$^{R24C}$, Cyclin D1, and TERT, is better able to preserve the original nature of dental pulp stem cells without chromosomal instability compared to other immortalization methods, such as the use of SV40. However, in the G-banding analysis, we noted chromosomal aberrations in two out of 20 mitotic cells. We previously showed that 6.5% of cells exhibited abnormal chromosome patterns, even in wild type pig embryonic fibroblasts [26]. Therefore, there is a possibility that the ~10% observed chromosome abnormalities could be an artifact caused by sample preparation for chromosome analysis. On the other hand, immortalization driven by the expression of TERT alone was recently reported in cell types related to the dental field, including dental pulp cells [28,29], human deciduous tooth dental pulp cells [30], and gingival fibroblasts [31]. Although we also tried to obtain immortalized dental pulp stem cells through the expression of TERT alone, we were unable to, even after confirming the successful integration of TERT into cells by PCR. Dental pulp stem cells expressing TERT alone stopped proliferating soon after expanding into new plates for further analysis assays, whereas K4D and K4DT cells showed constant proliferation. We used the pCLXSH-TERT plasmid that utilizes

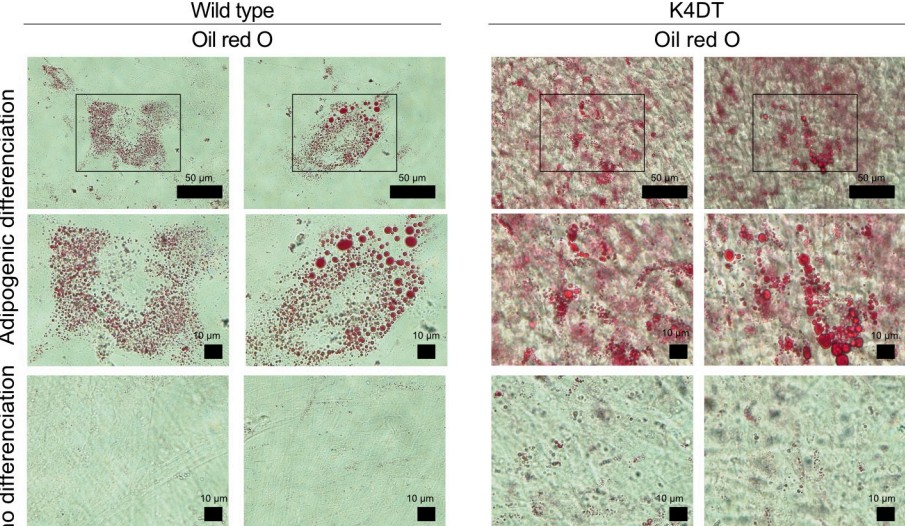

**Fig 9. Adipogenic differentiation properties of wild type and K4DT cells.** Representative Oil Red O staining of wild type and K4DT cells. For the Oil Red O staining, cells were cultured in adipoinduction medium for 23 days. Middle panels show high magnification of the boxed regions in upper panels.

the viral LTR promoter for TERT expression, whereas others reported that immortalized dental pulp cells using TERT could be generated using the pQCXIP plasmid [32]. There is a possibility that dental pulp stem cells expressing TERT alone using the CMV promoter, which is stronger than the LTR promoter, or changing culture conditions that stimulate the stress signaling pathway [33] might promote continuous proliferation in cells transformed with TERT alone.

We performed flow cytometric analysis and the differentiation experiments used as methods for the identification and characterization of dental pulp stem cells [34], and observed our established immortalized human dental pulp stem cells, K4DT cells maintained its original stemness characteristics without affecting differentiation potential. The results suggest that co-expression of CDK4$^{R24C}$, Cyclin D1, and TERT retain the original differentiation abilities of human dental pulp cells even though the cell proliferation was accelerated. Shiomi et al. reported that the co-expression of these three genes keeps the original cellular differentiation ability of human myogenic cells [17]. Our present results in differentiation experiments exhibited higher differentiation potential in the K4DT cells compared to that in primary human dental pulp stem cells, termed WT cells for both osteogenesis and adipogenesis. In these experiments, we used the WT cells at passage 8, which is passage number the cells have exhibited slower rates of cell proliferation as shown in Fig 4A. Therefore, the exact mechanism of the higher differentiation potential in K4DT cells has been unknown, the differentiation potential in K4DT cells might be due to keep the differentiation-inducing condition as primary human dental pulp stem cells in the early passages.

## Conclusion

We established novel human dental pulp stem cell lines without major chromosomal abnormalities or a transformed phenotype and losing their original characteristics by co-expressing mutant CDK4, Cyclin D1, and TERT. Our experimental results will contribute to the reduction of the therapeutic cost of pulp regeneration.

## Supporting information

**S1 Fig. No crop gel images of the PCR amplifications of Fig 3A.** The corresponding area of the gel images were indicated by white rectangle.
(TIFF)

**S2 Fig. No crop blot images of the western blots of Fig 3B.** The corresponding area of the gel images were indicated by white rectangle.
(TIFF)

## Acknowledgments

We would like to thank Dr. Akira Ishisaki (Iwate Medical University) for his technical help and fruitful advice.

## Author Contributions

**Conceptualization:** Tohru Kiyono, Tomokazu Fukuda.

**Data curation:** Ai Orimoto, Seiko Kyakumoto, Takahiro Eitsuka, Kiyotaka Nakagawa, Tohru Kiyono, Tomokazu Fukuda.

**Formal analysis:** Ai Orimoto.

**Investigation:** Ai Orimoto.

**Project administration:** Ai Orimoto.

**Supervision:** Tomokazu Fukuda.

**Validation:** Ai Orimoto, Tomokazu Fukuda.

**Writing – original draft:** Ai Orimoto.

**Writing – review & editing:** Tohru Kiyono, Tomokazu Fukuda.

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
