## [Decision Letter · Decision Letter 0]

7 Jan 2020

PONE-D-19-34813

Efficient immortalization of human dental pulp stem cells with expression of cell cycle regulators with intact chromosomal condition

PLOS ONE

Dear Orimoto,

Thank you for submitting your manuscript to PLOS ONE. After careful consideration, we feel that it has merit but does not fully meet PLOS ONE’s publication criteria as it currently stands. Therefore, we invite you to submit a revised version of the manuscript that addresses the points raised during the review process.

This study is of potential interest. but preliminary. As the reviewers specified the Authors must prove that cells do not die for senescence and keep their differentiating properties. To do so they must prove that after several passage they express antigens and are alive as well as that they are still capable to differentiate into osteogenic and adipogenic lineages.

We would appreciate receiving your revised manuscript by Feb 21 2020 11:59PM. To enhance the reproducibility of your results, we recommend that if applicable you deposit your laboratory protocols in protocols.io, where a protocol can be assigned its own identifier (DOI) such that it can be cited independently in the future. For instructions see: http://journals.plos.org/plosone/s/submission-guidelines#loc-laboratory-protocols

We look forward to receiving your revised manuscript.

Kind regards,

Gianpaolo Papaccio, M.D., Ph.D.

Academic Editor

PLOS ONE

Journal Requirements:

5. Please ensure that you refer to Figure 5 in your text as, if accepted, production will need this reference to link the reader to the figure.

Reviewers' comments:

Reviewer's Responses to Questions

**Comments to the Author**

1. Is the manuscript technically sound, and do the data support the conclusions?

Reviewer #1: No

Reviewer #2: Partly

2. Has the statistical analysis been performed appropriately and rigorously? 

Reviewer #1: Yes

Reviewer #2: I Don't Know

3. Have the authors made all data underlying the findings in their manuscript fully available?

Reviewer #1: Yes

Reviewer #2: Yes

4. Is the manuscript presented in an intelligible fashion and written in standard English?

Reviewer #1: Yes

Reviewer #2: Yes

5. Review Comments to the Author

Reviewer #1: In this paper authors established a novel immortalized dental pulp stem cell line by co-expressing a mutant cyclin-dependent kinase 4 (CDK4 R24C ), Cyclin D1, and telomerase reverse transcriptase (TERT). They concluded that the established K4DT cell line has the advantage of being easy to handle, resulting useful in dental pulp regeneration therapy.

The concept is interesting, but it represents only a preliminary study. The authors should test the characteristic of the new generated cell line, such as the stemness and the ability to differentiate at least in osteogenic lineage. For these reasons, the conclusions are not supported by data.

Reviewer #2: In this manuscript the Authors describe a new immortalized cell line obtained from dental pulp stem cells infected with retrovirus carrying the genes for a mutant cyclin-dependent kinase 4 (CDK4R24C), Cyclin D1, and telomerase reverse transcriptase (TERT). The Authors showed that such cell line have a normal karyotype and do not go in senescence after few passages. The Authors propose this cell line as a new tool to study dental pulp stem cells based regeneration protocols. The work is interesting and the experiments are clearly performed, however there are few concerns that need to be addressed a s follow:

In order to be used as a tool for regeneration experiments it not sufficient to prove that the cells do not go in senescence, the author should also prove that the cells keep their differentiation properties, so they should allow the cells to grow for few passages and then show that they are still able to differentiate toward at least osteogenic and adipogenic phenotype (just as a proof of concept).

Dental pulp stem cells have been described to express different markers (see as an example “Methods for the identification, characterization and banking of human DPSCs: current strategies and perspectives. Stem Cell Rev Rep. 2011 Sep;7(3):608-15. doi: 10.1007/s12015-011-9235-9.) the Author should show that the expression of stemness associated markers is not lost after passages.

6. PLOS authors have the option to publish the peer review history of their article (what does this mean?). If published, this will include your full peer review and any attached files.

Reviewer #1: No

Reviewer #2: No

---

## [Author Response · Author response to Decision Letter 0]

10 Feb 2020

Revision summary

We would like to submit our revised version of our manuscript, which entitled “Efficient immortalization of human dental pulp stem cells with expression of cell cycle regulators with the intact chromosomal condition”. All comments obtained from the reviewers were quite productive to improve our manuscript. We will list all comments and corresponding change listed below with point by point style. Furthermore, we will submit our revised manuscript with changes highlighted. In this round of revisions, we performed flow cytometric analysis and the differentiation experiments to test the characteristic of our generated cell line. We showed that our established immortalized human dental pulp stem cells preserved the expression of stemness associated markers and their differentiation properties even after passages. As we carried out our best efforts to satisfy the comments, we are glad to finish up the additional task, if it is essential for the acceptance of the manuscript.

Journal Requirements:

We performed double-check the formatting style of the PLoS One, and re-formatted the entire manuscript. Now, we hope that our modification of the manuscript style would be satisfactory to the comments.

 Based on the instruction, we included all experimental data used to the conclusions drawn in our manuscript. The data upload would be satisfactory to the comments from the journal office. 

 Based on the instruction, we uploaded full range blots and gel images as Supplementary Figures. 

Based on the instruction, the corresponding author create a new ORCID iD. 

5. Please ensure that you refer to Figure 5 in your text as, if accepted, production will need this reference to link the reader to the figure.

Based on the instruction, we included Figure 5 in our manuscript to allow the reader to link the figure. Thank you for the suggestion.

Review Comments to the Author

Reviewer #1: In this paper authors established a novel immortalized dental pulp stem cell line by co-expressing a mutant cyclin-dependent kinase 4 (CDK4 R24C ), Cyclin D1, and telomerase reverse transcriptase (TERT). They concluded that the established K4DT cell line has the advantage of being easy to handle, resulting useful in dental pulp regeneration therapy.

The concept is interesting, but it represents only a preliminary study. The authors should test the characteristic of the new generated cell line, such as the stemness and the ability to differentiate at least in osteogenic lineage. For these reasons, the conclusions are not supported by data.

We appreciate very much to summarize the outline of our publication. Based on the comments, to test the characteristic of our generated cell line, termed K4DT cells, we performed flow cytometric analysis and the differentiation experiments. As a result, flow cytometry analysis of K4DT cells showed the high expression of the common stem cell markers in mesenchymal stem cell, CD90, and low expression of hematopoietic markers, CD34 and CD45, similar to the results observed in wild type cells. In the differentiation experiments, we tested the ability to differentiate in the osteogenic and adipogenic lineage. The K4DT cells were differentiated into osteoblasts immunostained with anti-osteocalcin antibody and showed marked mineralization by Alizarin Red S staining. K4DT cells also differentiated adipocytes and showed positive staining of numerous oil droplets by Oil red O.

Our data suggest that our established cell line maintained its original stemness characteristics without affecting differentiation potential. 

To report the results described above, we involved new figures (Figure 7, 8, and 9 in the revised manuscript) and corresponding descriptions in the section of Materials and Methods (lines 157-183), Results (lines 248-268), and Discussion (lines 323-336).

Reviewer #2: In this manuscript the Authors describe a new immortalized cell line obtained from dental pulp stem cells infected with retrovirus carrying the genes for a mutant cyclin-dependent kinase 4 (CDK4R24C), Cyclin D1, and telomerase reverse transcriptase (TERT). The Authors showed that such cell line have a normal karyotype and do not go in senescence after few passages. The Authors propose this cell line as a new tool to study dental pulp stem cells based regeneration protocols. The work is interesting and the experiments are clearly performed, however there are few concerns that need to be addressed a s follow:

In order to be used as a tool for regeneration experiments it not sufficient to prove that the cells do not go in senescence, the author should also prove that the cells keep their differentiation properties, so they should allow the cells to grow for few passages and then show that they are still able to differentiate toward at least osteogenic and adipogenic phenotype (just as a proof of concept).

Dental pulp stem cells have been described to express different markers (see as an example “Methods for the identification, characterization and banking of human DPSCs: current strategies and perspectives. Stem Cell Rev Rep. 2011 Sep;7(3):608-15. doi: 10.1007/s12015-011-9235-9.) the Author should show that the expression of stemness associated markers is not lost after passages. 

We appreciate very much to summarize the outline of our publication and for the critical and productive comments to our publication. Based on the suggestion, to determine whether our immortalized cells, termed K4DT cells, keep the original characteristic of dental pulp stem cells, the stemness and their differentiation abilities were assessed by using the K4DT cells at passage 8, 15, or 16. As described in the responses to the Reviewer’s comment #1, flow cytometry analysis of K4DT cells at passage 8 showed the high expression of the common stem cell markers in mesenchymal stem cell, CD90 and low expression of hematopoietic markers, CD34 and CD45, similar to the results observed in wild type cells. In addition, we studied their differentiation abilities for osteogenesis and adipogenesis by using the K4DT cells at passage15 and 16. As a result, we observed that the K4DT cells differentiated into osteoblasts and adipocytes in each differentiation medium containing differentiation inducers. K4DT cells at passages 15 after approximately 3 months that started cell culture showed positive staining of calcified materials by Alizarin red S after 17 days. The K4DT cells also differentiated into osteoblasts immunostained with anti-osteocalcin antibody after 25 days. After 23 days of culture with adipoinduction medium, the K4DT at passages 16 developed numerous lipid droplets stained with Oil Red O. These results indicated that our established immortalized human dental pulp stem cells preserved the expression of stemness associated markers and their differentiation properties even after passages. 

To report the results described above, we involved new figures (Figure 7, 8, and 9 in the revised manuscript) and corresponding descriptions in the section of Materials and Methods (lines 157-183), Results (lines 248-268), and Discussion (lines 323-336).

---

## [Decision Letter · Decision Letter 1]

19 Feb 2020

PONE-D-19-34813R1

Efficient immortalization of human dental pulp stem cells with expression of cell cycle regulators with the intact chromosomal condition

PLOS ONE

Dear Orimoto,

Thank you for submitting your manuscript to PLOS ONE. After careful consideration, we feel that it has merit but does not fully meet PLOS ONE’s publication criteria as it currently stands. Therefore, we invite you to submit a revised version of the manuscript that addresses the points raised during the review process.

The paper has been improved but requires some minor amendments as specified below.

We would appreciate receiving your revised manuscript by Apr 04 2020 11:59PM. To enhance the reproducibility of your results, we recommend that if applicable you deposit your laboratory protocols in protocols.io, where a protocol can be assigned its own identifier (DOI) such that it can be cited independently in the future. For instructions see: http://journals.plos.org/plosone/s/submission-guidelines#loc-laboratory-protocols

We look forward to receiving your revised manuscript.

Kind regards,

Gianpaolo Papaccio, M.D., Ph.D.

Academic Editor

PLOS ONE

Additional Editor Comments (if provided):

The manuscript is partially improved. Actually the Authors must add either in the Introduction and in the Discussion Sections some paragraphs regarding the osteogenic differentiation that DPSCs spontaneously undergo, citing the following previous literature: Clin Sci. 131, Issue 8, 2017, Pages 699-713 regarding the capability of DPSCs to build a human bone tissue as well as J cell Physiol 2013, 228, pp 1149/53 regarding osteocalcin.

Moreover they must add some info regarding the methods previously raised citing Stem Cell Rev rep 2011, 7: 608/15.

Reviewers' comments:

Reviewer's Responses to Questions

**Comments to the Author**

1. If the authors have adequately addressed your comments raised in a previous round of review and you feel that this manuscript is now acceptable for publication, you may indicate that here to bypass the “Comments to the Author” section, enter your conflict of interest statement in the “Confidential to Editor” section, and submit your "Accept" recommendation.

Reviewer #1: All comments have been addressed

Reviewer #2: All comments have been addressed

2. Is the manuscript technically sound, and do the data support the conclusions?

Reviewer #1: (No Response)

Reviewer #2: Yes

3. Has the statistical analysis been performed appropriately and rigorously? 

Reviewer #1: (No Response)

Reviewer #2: I Don't Know

4. Have the authors made all data underlying the findings in their manuscript fully available?

Reviewer #1: (No Response)

Reviewer #2: Yes

5. Is the manuscript presented in an intelligible fashion and written in standard English?

Reviewer #1: (No Response)

Reviewer #2: Yes

6. Review Comments to the Author

Reviewer #1: (No Response)

Reviewer #2: The Authors have adressed the comments raised by this reviewer. The manuscriptresults greatly improved.

7. PLOS authors have the option to publish the peer review history of their article (what does this mean?). If published, this will include your full peer review and any attached files.

Reviewer #1: No

Reviewer #2: No

---

## [Author Response · Author response to Decision Letter 1]

19 Feb 2020

February 20, 2020

Editorial Office

PLOS ONE

Dear Professor Gianpaolo Papaccio

 We have attached our revised manuscript (PONE-D-19-34813R1) file entitled ‘: Efficient immortalization of human dental pulp stem cells with expression of cell cycle regulators with the intact chromosomal condition.’ We appreciate your invitation to submit a revised manuscript and have provided an improved manuscript. We also provide responses for Additional Editor Comments. Furthermore, we will submit our revised manuscript with changes highlighted. As we carried out our best efforts to satisfy the comments, we are happy to finish up the additional task, if it is essential for the acceptance of the manuscript.

Additional Editor Comments (if provided):

The manuscript is partially improved. Actually the Authors must add either in the Introduction and in the Discussion Sections some paragraphs regarding the osteogenic differentiation that DPSCs spontaneously undergo, citing the following previous literature: Clin Sci. 131, Issue 8, 2017, Pages 699-713 regarding the capability of DPSCs to build a human bone tissue as well as J cell Physiol 2013, 228, pp 1149/53 regarding osteocalcin.

We appreciate very much for the supportive comments to our publication. We agree with this comment that we must add these references. As suggested, we have added two references as suggested, and have improved the corresponding manuscript at line 49-51 and lines 261-262. 

Moreover they must add some info regarding the methods previously raised citing Stem Cell Rev rep 2011, 7: 608/15.

Thank you for your suggestion. As suggested, we included the reference in the section of Discussion and improved the corresponding manuscript at line 326-329.

 We hope that the revised manuscript is now acceptable for publication in Plos one. If you require any further information, please do not hesitate to contact me.

Sincerely,

Ai Orimoto

Ai Orimoto, DDS, Ph. D.

Graduate School of Science and Engineering, Iwate University, 

4-3-5, Ueda, Morioka, Iwate, 020-8551, Japan

TEL: 81-19-621-6375

E-mail: orimoto@ iwate-u.ac.jp

---

## [Editor Report · Decision Letter 2]

20 Feb 2020

Efficient immortalization of human dental pulp stem cells with expression of cell cycle regulators with the intact chromosomal condition

PONE-D-19-34813R2

Dear Dr. Orimoto,

We are pleased to inform you that your manuscript has been judged scientifically suitable for publication and will be formally accepted for publication once it complies with all outstanding technical requirements.

With kind regards,

Gianpaolo Papaccio, M.D., Ph.D.

Academic Editor

PLOS ONE

Additional Editor Comments (optional):

The Authors answered to all the previous comments.
---

## [Editor Report · Acceptance letter]

24 Feb 2020

PONE-D-19-34813R2 

Efficient immortalization of human dental pulp stem cells with expression of cell cycle regulators with the intact chromosomal condition 

Dear Dr. Orimoto:

I am pleased to inform you that your manuscript has been deemed suitable for publication in PLOS ONE. Congratulations! Your manuscript is now with our production department. 

With kind regards,

on behalf of

Prof. Gianpaolo Papaccio 

Academic Editor

PLOS ONE